# Off-Label Use of Crisdesalazine (GedaCure) in Meningoencephalitis in Two Dogs

**DOI:** 10.3390/vetsci10070438

**Published:** 2023-07-05

**Authors:** Saeyoung Lee, Woo-Jin Song, Jongjin Park, Minkun Kim, Sangkyung Choen, Myung-Chul Kim, Hyohoon Jeong, Youngmin Yun

**Affiliations:** 1Laboratory of Veterinary Internal Medicine, College of Veterinary Medicine, Jeju National University, Jeju 63243, Republic of Korea; qr3397@naver.com (S.L.); pjj4024@naver.com (J.P.); ketopi92@naver.com (M.K.); hjeong@jejunu.ac.kr (H.J.); dvmyun@jejunu.ac.kr (Y.Y.); 2The Research Institute of Veterinary Science, College of Veterinary Medicine, Jeju National University, Jeju 63243, Republic of Korea; mck@jejunu.ac.kr; 3Department of Surgical and Radiological Sciences, School of Veterinary Medicine, University of California, Davis, CA 95616, USA; chunsk1987@gmail.com; 4Diagnostic Laboratory Medicine, College of Veterinary Medicine, Jeju National University, Jeju 63243, Republic of Korea

**Keywords:** dog, meningoencephalitis, MUO, cytosine arabinoside, cyclosporine

## Abstract

**Simple Summary:**

Crisdesalazine is a multi-targeted drug with antioxidant and anti-inflammatory action to prevent amyloid plaque, neurofibrillary tangle, and neurodegeneration that cause dementia. We tried this drug on meningoencephalitis of unknown origin in dogs, expecting the improvement of their neurologic signs. In Case 1, the dog was tentatively diagnosed with meningoencephalitis of unknown origin by MRI and has started treatment with prednisolone and cytarabine. However, recurrent severe bacterial cystitis has occurred, suspected as the side effect of immunosuppressants. After adding crisdesalazine in ongoing therapy, the patient could reduce the dosage of the drugs successfully. In Case 2, the dog was tentatively diagnosed with meningoencephalitis of unknown origin through its history using clinical signs, blood analysis, neurologic examination, magnetic resonance imaging, and cerebrospinal fluid analysis. The dog has started treatment with cytarabine, prednisolone, and levetiracetam and the alleviation of the clinical signs was inadequate. However, after adding crisdesalazine in ongoing treatment, the neurologic signs of the dog were markedly improved. To the best of our knowledge, no previous report has documented clinical cases of reduced neurological signs in dogs with meningoencephalitis of unknown origin after adding crisdesalazine as an adjunct to ongoing immunosuppressive treatment.

**Abstract:**

An 8-year-old, castrated male Shih-tzu dog (Case 1) showing ataxia and gait disorder was referred for neurological examination and magnetic resonance imaging. Through comprehensive examinations, the patient was tentatively diagnosed with meningoencephalitis of unknown origin (MUO) and treatment with prednisolone and cytosine arabinoside was started. The symptoms were improving with immunosuppressive treatment. However, severe bacterial cystitis occurred and we could not avoid tapering off prednisolone. Then, neurological signs recurred. Therefore, we added crisdesalazine, which allowed us to reduce the daily dosage of immunosuppressants easily. In another case, a 4-year-old, spayed female Yorkshire terrier dog (Case 2) was referred to our hospital showing a head tilt, circling, and loss of the menace reflex. The patient was tentatively diagnosed with MUO and treatment with some immunosuppressants was attempted. The clinical symptoms improved, but the alleviation was inadequate. Thus, we added crisdesalazine. The neurological signs then markedly improved. Moreover, the drugs could be tapered off more easily than before. Crisdesalazine is a novel drug that has antioxidant and anti-inflammatory action in brain disease and is used particularly for dementia. In this paper, we tried an off-label use of this drug in canine MUO patients, and found that it had, in these two patients, additional therapeutic effects on the MUO.

## 1. Introduction

Canine meningoencephalitis (ME) can be classified into granulomatous meningoencephalomyelitis, necrotizing meningoencephalitis, and necrotizing leukoencephalitis [1]. These are common inflammatory conditions of the canine central nervous system. An exact diagnosis can be made through postmortem autopsy with histopathology [1]. The antemortem tentative diagnosis is made based on magnetic resonance imaging (MRI) findings and cerebrospinal fluid (CSF) analysis [2]. If no source of infection is detected in the CSF, the term ME of unknown etiology/origin (MUE/MUO) is used to make a tentative diagnosis [1,2,3].

MUO is currently typically treated with immunosuppressive drugs [3], such as a combination of glucocorticoids and cytosine arabinoside or cyclosporine [4,5], and median survival time ranges from 26 to 1834 days [2,3]. Despite of these managements using immunosuppressants, MUO is progressive in most cases [2,3,6].

In this case report, we used crisdesalazine (AAD-2004, GedaCure; Yuhan Pharmaceuticals, Seoul, Republic of Korea), a drug with anti-inflammatory and antioxidant effects, as an off-label indication to treat two dogs with MUO under ongoing immunosuppressive treatment and found that it had additional beneficial effects in this condition.

## 2. Case Description

### 2.1. Dog 1

An 8-year-old, 6.7 kg, castrated male Shih Tzu dog was referred to our veterinary medical teaching hospital with acute tremor and ataxia. On physical examination, the vital signs were normal including systolic blood pressure (130 mmHg, Doppler method). During neurological examination, the patient showed left head tilt and turn, clonic tremor, and lumbar pain. On neurological examination, the patient showed bilateral abnormalities on both cranial nerve and postural reflexes. On biochemical profiles, alkaline phosphatase was 285 U/L (reference interval (RI), 20–150 U/L), while other values, (blood glucose concentration [GLU], 103 mg/dL, RI, 74–143 mg/dL; blood urea nitrogen levels [BUN], 14 mg/dL, RI, 7–27 mg/dL; blood ammonia concentration [NH3], 72 µmol/L, RI, 16–75 µmol/L) were normal. Mild anemia was revealed on complete blood count (hematocrit, 35.4%, RI, 37–55%; white blood cells, 11.1 × 109/L, RI, 6–17 × 109/L).The C-reactive protein level (CRP) was 28.3 (RI, 0–20 mg/L). The potassium level was 2.8 mmol/L (RI, 3.5–5.8 mmol/L). Thus, the patient was hospitalized and treated for anemia and electrolyte imbalance. However, the neurological signs remained. There were no specific findings on thoracic and abdominal radiography or ultrasonography. Urine specific gravity (USG) was 1.019 (RI, 1.015–1.045 for normally hydrated dogs) and there were no remarkable findings on a urine dipstick test. After eliminating the extracranial causes of neurological symptoms, we concluded that the neurolocalisation was intracranial and an MRI was performed. Diffuse and multifocal T2/FLAIR hyperintense signals were observed in the cerebrum and hypothalamus, as well as the left brainstem bilaterally. In addition, mild contrast enhancement was detected (Figure 1A). On CSF examination, neutrophils appeared to be predominant with a few lymphocytes, and no pathogens were detected in the CSF using canine neurologic PCR panel (GreenVet, Yongin, Republic of Korea). Thus, the patient was tentatively diagnosed with MUO and was started on immunosuppressive treatment with cytosine arabinoside (cytarabine; 200 mg/m^2^ for 8 h, continuous rate infusion, every 3 weeks) and prednisolone 1 mg/kg per oral (PO) q 12 h on Day 2. The prednisolone was tapered off according to the previous study of immune-mediated disorder [7]. However, on Day 68 the dog showed pollakiuria and bacterial cystitis was diagnosed. Therefore, antibiotic treatment and periodic urinary tract flushing were performed. On Day 83, neurological signs recurred at a prednisolone dose of 0.75 mg/kg q 12 h; thus, the drug was increased to 1 mg/kg q 12 h and neurological signs improved again. Subsequently, prednisolone was retapered to 0.3 mg/kg q 12 h, at which point hindlimb ataxia and head tilt reappeared. Consequently, the drug was again increased to 0.7 mg/kg q 12 h on Day 186 (Figure 1B) and neurologic signs disappeared again.

Crisdesalazine (GedaCure) was added on Day 205 (with prednisolone 0.5 mg/kg q 12 h) as an anti-inflammatory drug, with the expectation of being able to reduce the immunosuppressant dosage. Before adding crisdesalazine, the dog could not have remained in remission with 1 mg/kg/day of prednisolone. However, after adding crisdesalazine as an adjunctive treatment, prednisolone could be tapered off to 0.25 mg/kg/day without relapse of neurological symptoms. In addition, the interval of cytarabine administration was increased from 4 to 9 weeks, whereas it was previously administered every 3 weeks (Figure 1B). Subsequently, side effects of immunosuppressants, such as bacterial cystitis, had not yet occurred (Day 300).

### 2.2. Dog 2

A 4-year-old, 1.7 kg, spayed female Yorkshire terrier dog was referred to the Veterinary Medical Teaching Hospital, Jeju National University, showing a head tilt, a head turn, and circling to the left, as well as barking into the air. On physical examination, the vital signs were normal, including systolic blood pressure (140 mmHg, Doppler method). During neurological examination, the patient had loss of bilateral menace reflexes and pupillary light responses and showed right positional nystagmus on cranial nerve reflexes. Moreover, the right-side reflexes were generally weak on postural reflexes. The biochemical profile was normal (GLU, 96 mg/dL, RI, 74–143 mg/dL; BUN, 19 mg/dL, RI, 7–27 mg/dL; NH3, 13 µmol/L, RI, 0–98 µmol/L). Complete blood count was also normal (hematocrit, 42.5%, RI, 37.3–61.7%; reticulocyte count, 46.2 K/μL, RI, 10–110 K/μL). CRP was 14.4 mg/L, within the reference range (0–20 mg/L). There were no specific findings on thoracic and abdominal radiography and ultrasonography. USG was 1.063, while protein 1 + level with pH 7 as revealed on urine dipstick. Therefore, we concluded that the neurolocalisation was intracranial and an MRI was performed. T2/FLAIR hyperintense signals were observed surrounding the left ventricles, in the lateral geniculate body, midbrain, and brainstem. Additionally, there was significant enlargement of the ventricles, particularly in the left ventricle. In addition, moderate contrast enhancement was detected (Figure 2A). Thus, the patient was tentatively diagnosed with MUO with suspected obstructive hydrocephalus due to MUO. CSF analysis could not be performed because of the owner’s refusal. As infection was not ruled out, only supportive medications (levetiracetam [20 mg/kg PO q 8 h], and mannitol [1 g/kg for 30 min, continuous rate infusion, pro re nata]) were prescribed [8,9]. Neurological signs did not improve; therefore, immunosuppressive drugs were administered with the owner’s approval. Mycophenolate mofetil (MMF) was first trialed along with prednisolone 1 mg/kg q 12 h (Day 0), but there was no remarkable improvement in the neurological signs. Therefore, MMF was changed to cytarabine (200 mg/m^2^ for 8 h, continuous rate infusion, every 3 weeks) on Day 48. The neurological signs then partially improved. Thus, the dosage of prednisolone was tapered off according to the previous study of immune-mediated disorder [7], however, neurological signs recurred at a prednisolone dose of 0.75 mg/kg q 12 h. Consequently, the drug was again increased to 1 mg/kg q 12 h on Day 152. The symptoms then improved, and the drug was again tapered off.

Crisdesalazine (GedaCure) was added on Day 188 with prednisolone 0.75 mg/kg q 12 h, under the expectation of achieving further anti-inflammatory effects. On the next day, the symptom of barking into the air ceased. Two weeks later (Day 202), both the head tilt and the distance of linear walking markedly improved (Appendix A). Moreover, neurological symptoms were no longer aggravated at the previous prednisolone threshold. To date (Day 280), the patient’s status is well-controlled at the same dose of prednisolone (Figure 2B).

## 3. Discussion

Crisdesalazine, which is 2-hydroxy-5-[2-(4-trifluoromethylphenyl)-ethylaminobenzoic acid] (AAD-2004), is a dual-function drug derived from aspirin and sulfasalazine [10], which are widely used to treat inflammatory diseases [11]. The drug has an anti-oxidative property as a potent spin trapping molecule and also has an anti-inflammatory effect as a microsomal prostaglandin E synthase-1 (mPGES-1) inhibitor [10]. In a mouse model with amyotrophic lateral sclerosis (ALS), AAD-2004 significantly delayed disease onset and extended survival, and also showed better safety and resulted in longer survival than that with conventional non-steroidal anti-inflammatory drugs (NSAIDs) such as riluzole or ibuprofen [10]. Unlike NSAIDs, which cause gastric damage at therapeutic doses, oral administration of 1000 mg/kg of AAD-2004, which was 400-fold higher than the maximal efficacy dose in SOD1-G93A mouse models, did not result in gastric bleeding [10]. The drug also showed neuroprotective effects on mouse models of Alzheimer’s disease, by blocking the increase in lipid peroxidation and suppressing the reactive oxygen species (ROS) levels in the brain [12]. Crisdesalazine (Gedacure^®^, GNT Pharma, Yongin, Republic of Korea) was approved by the Korean Animal and Plant Quarantine Agency in 2021 for management of dementia. Although this novel drug has not been approved for meningoencephalitis, it was tempting to speculate that its anti-inflammatory effects might have additional therapeutic effects in MUO dogs.

PGE2 is thought to be an important mediator of inflammation in peripheral tissues but recent studies have clearly shown the involvement of PGE2 in several inflammatory brain diseases in mice, rats, and humans [13]. PGE2 is the most common prostaglandin involved in the brain [13]. It plays a role in neuronal excitation and plasticity, which contributes to learning and memory, in neuronal proliferation and differentiation, and in dendritic spine formation [13]. Under pathological conditions, excess PGE2 is released in inflammatory brain lesion sites and contributes to symptom progression [13]. Although it is not confirmed in dogs, based on these findings, it is tempting to speculate that PGE2 is also involved in inflammatory brain disease in dogs.

In the PGE2 biosynthetic pathway, the prostaglandin phospholipase A2 releases arachidonic acid (AA) from membrane phospholipids. AA is metabolized to the endoperoxide prostaglandin H2 by cyclo-oxygenase (COX) and is then metabolized to PGE2 by PGES [13]. Among three major isoforms of PGES, only mPGES-1 is an inducible enzyme, which is upregulated by pro-inflammatory stimuli and in various models of inflammatory brain disease [13,14]. mPGES-1 is induced in the delayed phase of inflammation, which is involved in wakefulness, inflammation, hyperalgesia, pain, fever, and cancer [14]. Currently, traditional NSAIDs inhibiting both COX-1 and COX-2 activities are widely used in the treatment of rheumatoid arthritis, osteoarthritis, and inflammatory pain [15]. However, COX-1 inhibition has side effects on the gastrointestinal (GI) tract and selective COX-2 inhibitors can lead to thrombosis and cardiovascular side effects by reducing vasorelaxation [15]. Thus, an mPGES-1 inhibitor, such as AAD-2004, is a PGE2-specific anti-inflammatory drug that is an alternative to NSAIDs or selective COX-2 inhibitors, with minimal side effects.

Although the exact etiology and pathophysiology of MUO are currently unknown, the suspected triggers of the condition include environmental factors and antigens [1,2]. Considering the generally positive response to immunosuppressive treatment, it is thought that the conditions comprising MUO are immune-mediated [16,17]. In this study, we used crisdesalazine as an add-on to ongoing immunosuppressive treatment, with an expectation of anti-inflammatory and anti-oxidative benefits, or an immunosuppressant-sparing effect on this inflammatory brain disease.

Antioxidants are compounds that reduce ROS levels by interrupting the various steps of free radical-mediated oxidative processes [18]. The α, α-diphenyl-β-picrylhydrazyl (DPPH) free radical scavenging method is widely used to evaluate antioxidant activity [19]. In ALS mouse models, AAD-2004 rapidly reacted with DPPH, which is a stable free radical, and showed a much higher potency than vitamin E, sulfasalazine, salicylic acid, and aspirin, which suggests that AAD-2004 has higher anti-oxidative activity than other known antioxidants [19].

Electron spin resonance (ESR) is an indirect method for detecting free radicals, and some radicals have a very short relaxation time to be observed by ESR [20]. Spin trapping is the complementary method for overcoming this problem [20]. 5,5-Dimethyl-1-pyrroline-N-oxide (DMPO) is a spin-trapping agent, that reacts with HO and yields DMPO-OH [21]. Therefore, the ESR spectra of DMPO-OH indirectly expresses the levels of free radicals. In ALS mouse models, addition of AAD-2004 could reduce the levels of DMPO-OH, and virtually blocked the ESR-spectra of AAD-2004 as low as 50 nM [10]. Therefore, AAD-2004 reduces the levels of free radicals and could be a potent spin-trapping molecule, as an alternative of DMPO [10]. Based on these previous studies, the increased anti-oxidative effect obtained by adding AAD-2004 as treatment may have contributed to improvement of the neurological signs in these two cases.

It has been reported that general complications of long-term immunosuppressive therapy are an increased risk of infection and malignancy [21,22]. Specifically, the side effects of long-term glucocorticoid treatment include gastrointestinal bleeding, increased risk of pancreatitis, myopathy, skin insufficiency, ophthalmological diseases such as cataract and glaucoma, neuropsychiatric symptoms, and susceptibility to infection [7]. These side effects could be avoided or addressed by using corticosteroid-sparing therapies and tapering of corticosteroids to the minimal effective dose [21,22]. In case 1, bacterial cystitis occurred and repeated waxing-and-waning according to prednisolone tapering. However, after adding crisdesalazine (AAD-2004), prednisolone was tapered off more easily, without symptom recurrence. In case 2, neurological signs recurred at a certain dosage while tapering prednisolone, but it no longer recurred at that threshold dose, and the dog showed behavior improvement after adding crisdesalazine.

There are some limitations in this case report. First, only two cases were included in this report. Second, these dogs were not followed up for over 12 months. Further studies with prospective study design using larger samples are needed.

## 4. Conclusions

Our experience with these two cases demonstrated that crisdesalazine could be a novel therapeutic option as an add-on therapy that has antioxidant and anti-inflammatory effects in inflammatory brain conditions, pending further long-term trials and collecting safety data in MUO dogs.

## Figures and Tables

**Figure 1 vetsci-10-00438-f001:**
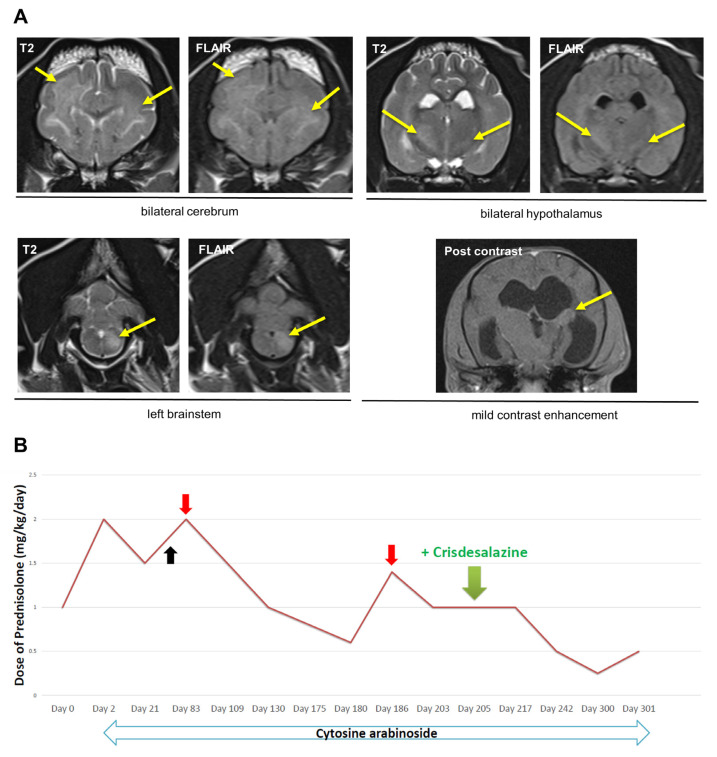
Diagnostic magnetic resonance imaging, and therapeutic course of immunosuppressants in Dog 1. (**A**) Diffuse and multifocal T2/FLAIR hyperintense signals (indicated by yellow arrows) were observed in the cerebrum and hypothalamus, as well as the left brainstem bilaterally. In addition, mild contrast enhancement was detected. These findings are consistent with meningoencephalitis. (**B**) After Day 205 (when crisdesalazine was added), dosages of prednisolone were successfully reduced without relapse of neurological signs. On Day 68 (black arrow) bacterial cystitis occurs. On Days 83 and 186 (red arrows), dosage of prednisolone is increased due to recurrence of neurologic signs.

**Figure 2 vetsci-10-00438-f002:**
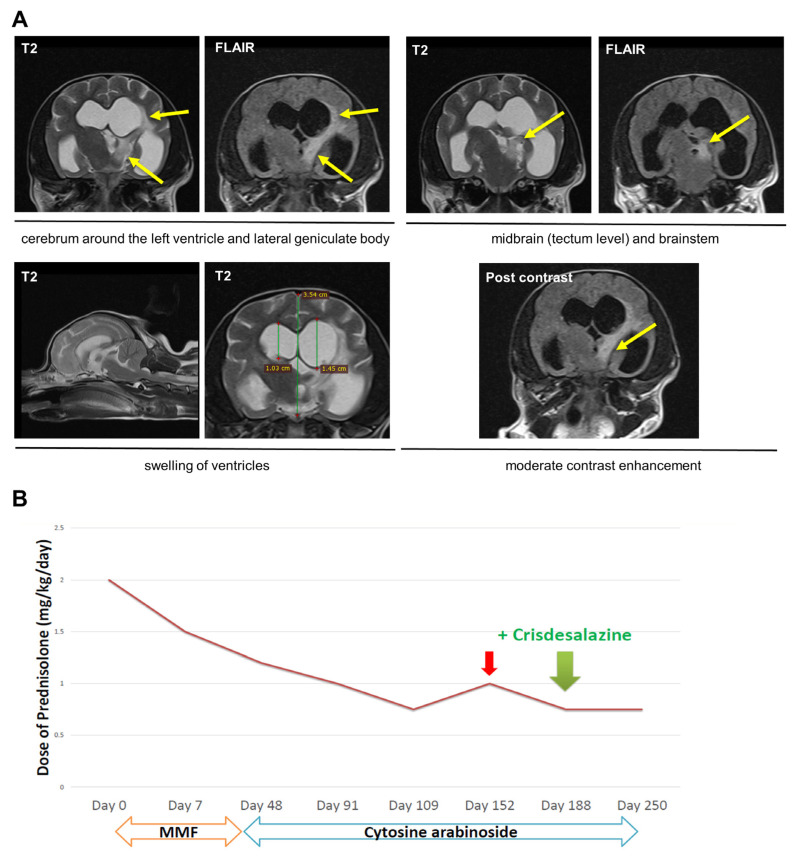
Diagnostic magnetic resonance imaging, and therapeutic course of immunosuppressants in Dog 2. (**A**) T2/FLAIR hyperintense signals (yellow arrows) are observed surrounding the left ventricles, in the lateral geniculate body, midbrain, and brainstem. Additionally, there is significant enlargement of the ventricles, particularly in the left ventricle. In addition, moderate contrast enhancement was detected. These findings are consistent with a diagnosis of meningoencephalitis with obstructive hydrocephalus. (**B**) After Day 188 (when crisdesalazine was added), neurological signs improved without increasing the dosages of immunosuppressants (cytosine arabinoside and prednisolone). On Day 152 (red arrow), dosage of prednisolone is increased due to recurrence of neurologic signs.

## Data Availability

The data presented in this study are contained within the article.

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
