# Peer review of "Off-Label Use of Crisdesalazine (GedaCure) in Meningoencephalitis in Two Dogs"

_vetsci, 2023, doi:10.3390/vetsci10070438_

Round 1

Reviewer 1 Report

Dear authors, 

This is a very relevant case report as we have to look at potential novel treatments for this devastating disease and I agree that sometimes trials will be made initially with off labels drugs for veterinary use. However, it is important to include all the information that is known about this drug in relation to dogs before making such a strong argument about the efficacy of the drug. do we know anything about the PK/PD , any other trials for other diseases? any information about biomarkers such as C reactive protein going back to normal? any blood tests conducted at follow up to show no side effects? consider adding a table with follow up blood tests to show no detrimental effects?

If none of this information is available, I would change the conclusion and report the novel usage of this drug as a potential add on therapy for MUE pending further trials and collecting safety data in dogs. 

The statement that all dogs will die due to MUE is not correct and you should rephrase it based a more thorough review articles in the field. There is a prior manuscript by Cornelis et al. that stated even 2250 days of survival and we know that some dogs recover fully. 

please avoid the term ME and MUE interchangeably. please refer to just MUE after the initial introduction to the topic. 

Dear authors,

I would recommend improving the medical terms in this report. some examples include: ventricles are not swollen, but rather : ventriculomegaly. 

The description of the images is lacking medical terms. please refer to Cornelis 2019 ( you cited this article) to describe the images in much more details. 

Instead of stating: bilateral hypothalamus, rephrase to : High T2/FLAIR signals were also detected in the  hypothalamus bilaterally and left brainstem levels.

line 87 states: " with a few lymphocytes,  and negative for all neurological infectious pathogens. "

This line is not clear, did you send the CSF for culture and sensitivity? or are you trying to say that no pathogens were noted in analysis of the CSF ? if so erase the words: " all neurological infectious " from the sentence as not all pathogens are noted in a regular CSF analysis. 

reference interval is repeated many times in the text instead of just once and (RI) after.

Author Response

We are very pleased to have been given another opportunity to revise our manuscript entitled “Off-label use of crisdesalazine (GedaCure) in meningoencephalitis in two dogs” for Veterinary Sciences. We want to extend our appreciation to you and the reviewers for taking the time and effort necessary to provide such insightful guidance. We have carefully considered comments offered by the reviewers. Herein, we explain how we revised the paper based on those comments and recommendations. The manuscript has certainly benefited from these revision suggestions. We look forward to working further with you and the reviewers to move this manuscript closer to publication.

Reveiwer #1
1. COMMENT: This is a very relevant case report as we have to look at potential novel treatments for this devastating disease and I agree that sometimes trials will be made initially with off labels drugs for veterinary use. However, it is important to include all the information that is known about this drug in relation to dogs before making such a strong argument about the efficacy of the drug. do we know anything about the PK/PD , any other trials for other diseases? any information about biomarkers such as C reactive protein going back to normal? any blood tests conducted at follow up to show no side effects? consider adding a table with follow up blood tests to show no detrimental effects?
If none of this information is available, I would change the conclusion and report the novel usage of this drug as a potential add on therapy for MUE pending further trials and collecting safety data in dogs. 
RESPONSE: Thank you for valuable comment, and we agree with you. Although crisdesalazine (GedacureⓇ, GNT Pharma, Yongin, Korea) was approved by the Korean Animal and Plant Quarantine Agency in 2021 for management of dementia, we could not show safety, PK/PD dats. Therefore, we revised the conclusion section as your comment. It has been described in the revision (Page 6, Line 184; Page 7, Lane 255). We hope our approach acceptable.

2. COMMENT: The statement that all dogs will die due to MUE is not correct and you should rephrase it based a more thorough review articles in the field. There is a prior manuscript by Cornelis et al. that stated even 2250 days of survival and we know that some dogs recover fully. 
RESPONSE: Thank you for your detailed comment. We agree with you. We modified the sentence as below: Despite of these managements using immunosuppressants, MUO is progressive in most cases (Cornelis, I.; Van Ham, L.; Gielen, I.; De Decker, S.; Bhatti, S.F.M. Clinical presentation, diagnostic findings, prognostic fac-tors, treatment and outcome in dogs with meningoencephalomyelitis of unknown origin: A review. Vet J. 2019, 244, 37–44; Jeffery N; Granger N. New insights into the treatment of meningoencephalomyelitis of unknown origin since 2009: A re-view of 671 cases. Front Vet Sci. 2023, 10, 1114798; Lawn RW; Harcourt-Brown TR. Risk factors for early death or euthanasia within 100 days of diagnosis in dogs with me-ningoencephalitis of unknown origin. Vet J. 2022, 287, 105884). It has been described in the revision (Page 2, Lane 58). We hope our approach acceptable.

3. COMMENT: please avoid the term ME and MUE interchangeably. please refer to just MUE after the initial introduction to the topic. 
RESPONSE: Thank you for your comment. We agree that it is better not to use the term MUE and ME interchangeably. Therefore we changed all the terms to MUO. We hope our approach acceptable.

Comments on the Quality of English Language
4. COMMENT: Dear authors,
I would recommend improving the medical terms in this report. some examples include: ventricles are not swollen, but rather : ventriculomegaly. 
The description of the images is lacking medical terms. please refer to Cornelis 2019 (you cited this article) to describe the images in much more details. 
Instead of stating: bilateral hypothalamus, rephrase to : High T2/FLAIR signals were also detected in the hypothalamus bilaterally and left brainstem levels.
RESPONSE: Thank you for your advice. We revised our description of the images in both cases. For case 1, Diffuse and multifocal T2/FLAIR hyperintense signals (frontal, parietal, and temporal lobes) are observed in the bilateral cerebrum and hypothalamus, as well as the left brainstem. In addition, mild contrast enhancement was detected; For case 2, T2/FLAIR hyperintense signals are observed surrounding the left ventricles, in the lateral geniculate body, midbrain, and brainstem. Additionally, there is significant enlargement of the ventricles, particularly in the left ventricle. In addition, moderate contrast enhancement was detected (Page2 , Line 84; Page4, Line 135). Also, we revised figure legends of Figure 1A and 2A.

5. COMMENT: line 87 states: " with a few lymphocytes,  and negative for all neurological infectious pathogens. "
This line is not clear, did you send the CSF for culture and sensitivity? or are you trying to say that no pathogens were noted in analysis of the CSF ? if so erase the words: " all neurological infectious " from the sentence as not all pathogens are noted in a regular CSF analysis. 
RESPONSE: Thank you for your detailed comment. In this case, On CSF examination, neutrophils appeared to be predominant with a few lymphocytes, and no pathogens were detected in the CSF using canine neurologic PCR panel (GreenVet, Yongin, Korea). Thus, the patient was tentatively diagnosed with multifocal MUO. It has been described in the revision (Page 2, Line 87). We hope our approach acceptable.

6. COMMENT: reference interval is repeated many times in the text instead of just once and (RI) after.
RESPONSE: Thank you for your comment. We changed all of them except first use.

Reviewer 2 Report

An interesting case report suggesting to use a novel drug to improve the outcomes in dogs with MUE.

Could you please provide more data why this novel drug has been chosen, is there evidence of it's use for meningoencephalitis in humans? Please add this information in the report.

What was the tapering down protocol used in both cases for the treatment of MUO? Was it based on any known literature?

The long-term outcome is not available in both cases, therefore, the efficacy of the novel medication is questionable.

Specific comments:

Please add to the text: what were the neurological localisations and main DDx in the described patients?

Post-contrast images are important for radiological diagnosis of MUE. Why the post-contrast images not performed in the described patients, or were they? Describe the images. 

Lines 83-84: 'in the bilateral cerebrum' change to 'in the cerebrum bilaterally'

Lines 85: 'bilateral hypothalamus' change to 'hypothalamus bilaterally'

Line 86: please describe CSF findings, was there any pleocytosis and/or increased protein count found?

Line 87: please specify which pathogens have been excluded, and was the blood and/or CSF bacterial culture performed as well in this case?

Line 91: please specify the clinical signs and work-up of cystitis in this dog. Also it was mentioned before that dog had USG of 1,019, which is not normal, please add comments on that (incl. suspected cause in this case).  

Line 94-95: please specify the tapering protocol: how fast and when exactly was prednisolone tapered?

Lines 96-97: did the increase of medication dose improve clinical signs

Line 108: how can you claim that crisdesalazine was preventing the relapse?

Line 132: where was the obstruction causing the suspected obstructive hydrocephalus localised based on the radiological MRI findings?

Line 134: please explain the rationale behind the use of levetiracetam in this case.

Line 140-143: please specify the tapering down protocol of prednisolone in this case as well

DISCUSSION

Lines 176-194: is there any proof available showing an involvement of PGE2 in canine MUE? If yes, please add the literature, if not, please add a discussion that it has not been confirmed in dogs etc.

Line 204-207: please revise if these median survival times are accurate and add more recent literature as reference 

Lines 212-240: these paragraphs need to be shortened to much less text in my opinion, including only main points and describing how it relates to the canine MUO patients and the rationale of using the novel drug as an add on in MUO cases.

Lines 241-246: rewrite it with discussion on main side effects in canine patients.

I suggest to add discussion that dog 2 was showing behavioural changes and they might have improved by using the crisdesalazine

Please add limitations of the study in the discussion section.

Author Response

We are very pleased to have been given another opportunity to revise our manuscript entitled “Off-label use of crisdesalazine (GedaCure) in meningoencephalitis in two dogs” for Veterinary Sciences. We want to extend our appreciation to you and the reviewers for taking the time and effort necessary to provide such insightful guidance. We have carefully considered comments offered by the reviewers. Herein, we explain how we revised the paper based on those comments and recommendations. The manuscript has certainly benefited from these revision suggestions. We look forward to working further with you and the reviewers to move this manuscript closer to publication.

Reveiwer #2
An interesting case report suggesting to use a novel drug to improve the outcomes in dogs with MUE.

1. COMMENT: Could you please provide more data why this novel drug has been chosen, is there evidence of it's use for meningoencephalitis in humans? Please add this information in the report.
RESPONSE: Thank you for valuable comment. Crisdesalazine (GedacureⓇ, GNT Pharma, Yongin, Korea) was approved by the Korean Animal and Plant Quarantine Agency in 2021 for management of dementia. Although this novel drug has not been approved for meningoencephalitis, it was tempting to speculate that its anti-inflammatory effects might have additional therapeutic effects in MUO dogs. It has been described in the revision (Page 6, Line 184). We hope our approach acceptable.

2. COMMENT: What was the tapering down protocol used in both cases for the treatment of MUO? Was it based on any known literature?
RESPONSE: Thank you for your detailed comment. For both cases, the prednisolone was tapered off according to the previous study of immune-mediated disorder (Swann JW; Garden OA; Fellman CL; et al. ACVIM consensus statement on the treatment of immune-mediated hemolytic anemia in dogs. J Vet Intern Med. 2019, 33, 1141-1172). It has been described in the revision (Page 2, Line 93; Page 4, Line 148).  

3. COMMENT: The long-term outcome is not available in both cases, therefore, the efficacy of the novel medication is questionable.
RESPONSE: Thank you for your valuable comment, and we agree with you. Although our clarification of the advantages of crisdesalazine in these two cases is that the drug contributed to tapering the dosage of prednisolone effectively, we have described the limitation in the discussion section as below: There are some limitations in this case report. First, only two cases were included in this report. Second, these dogs were not followed up for over 12 months. Further studies with prospective study design using larger samples are needed (Page 7, line 249). We hope our approach acceptable.

Specific comments:
4. COMMENT: Please add to the text: what were the neurological localisations and main DDx in the described patients?
RESPONSE: Thank you for valuable comment. In case 1, On neurological examination, the patient showed bilateral abnormalities on both cranial nerve and postural reflexes. In addition, the extracranial causes of neurological symptoms were eliminated, therefore, we tentatively localized lesions in the brain, and MRI was performed. In case 2, During neurological examination, the patient had loss of bilateral menace reflexes and pupillary light responses, and showed right positional nystagmus on cranial nerve reflexes. And, the extracranial causes of neurological symptoms were eliminated, therefore, we tentatively localized lesions in the brain, and MRI was performed. It has been described in the revision (Page 2, Line 67-84; Page 4, Line 125-134). 

5. COMMENT: Post-contrast images are important for radiological diagnosis of MUE. Why the post-contrast images not performed in the described patients, or were they? Describe the images.
RESPONSE: Thank you for your valuable comment. We have added post-contrast image and its description for both cases (Page 3, Page 5).

6. COMMENT: Lines 83-84: 'in the bilateral cerebrum' change to 'in the cerebrum bilaterally'
Lines 85: 'bilateral hypothalamus' change to 'hypothalamus bilaterally'
RESPONSE: Thank you for your detailed comment. We have revised the description of MRI images in the revision as follow: Diffuse and multifocal T2/FLAIR hyperintense signals are observed in the cerebrum and hypothalamus, as well as the left brainstem bilaterally. (Page 2, Line 84).  

7. COMMENT: Line 86: please describe CSF findings, was there any pleocytosis and/or increased protein count found?
Line 87: please specify which pathogens have been excluded, and was the blood and/or CSF bacterial culture performed as well in this case?
RESPONSE: Thank you for your detailed comment. On CSF examination, neutrophils appeared to be predominant with a few lymphocytes, and no pathogens were detected in the CSF using canine neurologic PCR panel (GreenVet, Yongin, Korea). It has been described in the revision (Page 2, Line 87). We hope our approach acceptable.

8. COMMENT: Line 91: please specify the clinical signs and work-up of cystitis in this dog. Also it was mentioned before that dog had USG of 1,019, which is not normal, please add comments on that (incl. suspected cause in this case). 
RESPONSE: Thank you for your detailed comment. The patient had showed pollakiuria, and it has been describe in the revision (Page 2, Line 94). And about the USG, we followed the reference ranges from International Renal Interest Society (IRIS) which says ‘normally hydrated individuals are often closer to 1.015 to 1.045 for dogs’. It has been described in the revision (Page 2, Line 81). We hope our approach acceptable.

9. COMMENT: Line 94-95: please specify the tapering protocol: how fast and when exactly was prednisolone tapered?
RESPONSE: Thank you for detailed comment. For both cases, the prednisolone was tapered off according to the previous study of immune-mediated disorder (Swann JW; Garden OA; Fellman CL; et al. ACVIM consensus statement on the treatment of immune-mediated hemolytic anemia in dogs. J Vet Intern Med. 2019, 33, 1141-1172). It has been described in the revision (Page 2, Line 93; Page 4, Line 148). We hope our approach acceptable.

10. COMMENT: Lines 96-97: did the increase of medication dose improve clinical signs
RESPONSE: Thank you for detailed comment. Consequently, the drug was re-increased to 0.7 mg/kg q 12 h on Day 186 (Figure 1B) and neurologic signs disappeared again. It has been described in the revision (Page 2, Line 98).

11. COMMENT: Line 108: how can you claim that crisdesalazine was preventing the relapse?
RESPONSE: Thank you for valuable comment. Before adding crisdesalazine, neurologic signs of the dog could not reach complete remission with 1 mg/kg/day of prednisolone. However, after adding crisdesalazine as an ad-junctive treatment, prednisolone could be tapered off to 0.25 mg/kg/day without relapse of neurological symptoms. In addition, the interval of cytarabine administration was in-creased from 4 to 9 weeks, whereas it was previously administered every 3 weeks. Subsequently, side effects of immunosuppressants, such as bacterial cystitis, have not yet occurred until day 300. It has been described in the revision (Page 3, Line 114). We hope our approach acceptable.

12. COMMENT: Line 132: where was the obstruction causing the suspected obstructive hydrocephalus localised based on the radiological MRI findings?
RESPONSE: Thank you for your detailed comment. We have revised the description of MRI images as follow: Therefore, we tentatively localized lesions in the brain, and MRI was performed. T2/FLAIR hyperintense signals are observed surrounding the left ventricles, in the lateral geniculate body, midbrain, and brainstem. Additionally, there is significant enlargement of the ven-tricles, particularly in the left ventricle. It has been described in the revision (Page 4, Line 134). We hope our approach acceptable.

13. COMMENT: Line 134: please explain the rationale behind the use of levetiracetam in this case.
RESPONSE: Thank you for detailed comment. We added 2 consensus statements for references (Bhatti, S.F.; De Risio L.; Muñana, K. et al. International Veterinary Epilepsy Task Force consensus proposal: medical treatment of canine epilepsy in Europe. BMC Vet Res. 2015, 11, 176; Podell M; Volk HA; Berendt M; Löscher W; Muñana K; Patterson EE; Platt SR. 2015 ACVIM Small Animal Consensus Statement on Seizure Management in Dogs. J Vet Intern Med. 2016, 30, 477-90). We hope our approach acceptable.

14. COMMENT: Line 140-143: please specify the tapering down protocol of prednisolone in this case as well
RESPONSE: Thank you for detailed comment. . For both cases, the prednisolone was tapered off according to the previous study of immune-mediated disorder (Swann JW; Garden OA; Fellman CL; et al. ACVIM consensus statement on the treatment of immune-mediated hemolytic anemia in dogs. J Vet Intern Med. 2019, 33, 1141-1172). It has been described in the revision (Page 2, Line 93; Page 4, Line 148). We hope our approach acceptable.

DISCUSSION
15. COMMENT: Lines 176-194: is there any proof available showing an involvement of PGE2 in canine MUE? If yes, please add the literature, if not, please add a discussion that it has not been confirmed in dogs etc.
RESPONSE: Thank you for your valuable comment. As your comment, PGE2 is thought to be an important mediator of inflammation in peripheral tissues, but recent studies have clearly shown the involvement of PGE2 in several inflammatory brain disease only in mice, rats, and human. Although it is not confirmed in dogs, based on these findings, it is tempting to speculate that PGE2 is also involved in inflammatory brain disease in dogs. It has been described in the revision (Page 6, Line 189). We hope our approach acceptable.

16. COMMENT: Line 204-207: please revise if these median survival times are accurate and add more recent literature as reference 
RESPONSE: Thank you for your detailed comment. MUO is currently typically treated with immunosuppressive drugs (Jeffery N; Granger N. New insights into the treatment of meningoencephalomyelitis of unknown origin since 2009: A re-view of 671 cases. Front Vet Sci. 2023, 10, 1114798), such as a combination of glucocorticoids and cytosine arabinoside or cyclosporine (Brady SL; Woodward AP; le Chevoir M. Survival time and relapse in dogs with meningoencephalomyelitis of unknown origin treated with prednisolone and ciclosporin: a retrospective study. Aust Vet J. 2020, 98, 491-498; Barber R; Downey Koos L. Treatment With Cytarabine at Initiation of Therapy With Cyclosporine and Glucocorticoids for Dogs With Meningoencephalomyelitis of Unknown Origin Is Not Associated With Improved Outcomes. Front Vet Sci. 2022, 9, 925774), and me-dian survival time ranges from 26 to 1,834 days (Cornelis, I.; Van Ham, L.; Gielen, I.; De Decker, S.; Bhatti, S.F.M. Clinical presentation, diagnostic findings, prognostic fac-tors, treatment and outcome in dogs with meningoencephalomyelitis of unknown origin: A review. Vet J. 2019, 244, 37–44; Jeffery N; Granger N. New insights into the treatment of meningoencephalomyelitis of unknown origin since 2009: A re-view of 671 cases. Front Vet Sci. 2023, 10, 1114798). Despite of these managements using immunosuppressants, MUO is progressive in most cases (Cornelis, I.; Van Ham, L.; Gielen, I.; De Decker, S.; Bhatti, S.F.M. Clinical presentation, diagnostic findings, prognostic fac-tors, treatment and outcome in dogs with meningoencephalomyelitis of unknown origin: A review. Vet J. 2019, 244, 37–44; Jeffery N; Granger N. New insights into the treatment of meningoencephalomyelitis of unknown origin since 2009: A re-view of 671 cases. Front Vet Sci. 2023, 10, 1114798; Lawn RW; Harcourt-Brown TR. Risk factors for early death or euthanasia within 100 days of diagnosis in dogs with me-ningoencephalitis of unknown origin. Vet J. 2022, 287, 105884). It has been described in the revision (Page 2, Line 57). We hope our approach acceptable.

17. COMMENT: Lines 212-240: these paragraphs need to be shortened to much less text in my opinion, including only main points and describing how it relates to the canine MUO patients and the rationale of using the novel drug as an add on in MUO cases.
RESPONSE: Thank you for your detailed comment, and we agree with you. We have revised the discussion section as your comments.

18. COMMENT: Lines 241-246: rewrite it with discussion on main side effects in canine patients.
RESPONSE: Thank you for your detailed comment, and we have added the following references: 1) Swann JW; Garden OA; Fellman CL; et al. ACVIM consensus statement on the treatment of immune-mediated hemolytic anemia in dogs. J Vet Intern Med. 2019, 33, 1141-1172. DOI:10.1111/jvim.15463; 2) Günther C; Steffen F; Alder DS; Beatrice L; Geigy C; Beckmann K. Evaluating the use of cytosine arabinoside for treatment for recurrent canine steroid-responsive meningitis-arteritis. Vet Rec. 2020, 187, e7. DOI:10.1136/vr.105683; 3) Fukushima, K; Lappin, M; Legare, M; Veir, J. A retrospective study of adverse effects of mycophenolate mofetil administration to dogs with immune-mediated disease. J Vet Intern Med. 2021, 35, 2215– 2221. DOI:10.1111/jvim.16209. It has been described in the revision (Page 6, Line 236). 

19. COMMENT: I suggest to add discussion that dog 2 was showing behavioural changes and they might have improved by using the crisdesalazine.
RESPONSE: Thank you for detailed comment. We have added the sentence as follow: In case 2, neurological signs recurred at a certain dosage while tapering prednisolone, but it no longer recurred at that threshold dose, and the dog showed behavior improvement after adding crisdesalazine. It has been described in the revision (Page 7, Line 247).

20. COMMENT: Please add limitations of the study in the discussion section.
RESPONSE: Thank you for valuable comment. There are some limitations in this case report. First, only two cases were included in this report. Second, these dogs were not followed up for over 12 months. Further studies with prospective study design using larger samples are needed. It has been described in the revision (Page 7, Line 249). We hope our approach acceptable.

Reviewer 3 Report

Dear authors,

The topic itself is interesting as there is constant debate on what might be the best treatment. You may have overlooked it but a recent paper address this:

·      Jeffery N, Granger N. New insights into the treatment of meningoencephalomyelitis of unknown origin since 2009: A review of 671 cases. Front Vet Sci. 2023 Mar 15;10:1114798. doi: 10.3389/fvets.2023.1114798. PMID: 37008358; PMCID: PMC10050685.

I advise to in cooperate this reference as well.

Remarks:

1. The first remark I have that you should use one abbreviation to describe your patients. They are suffering from a Meningoencephalo(myelitis) of Unknown Origin and the abbreviation is MUO. In your paper you constantly mix the abbreviations (f.i.line 28, 32). As you are talking about dogs with MUO stick to MUO.

2. Make clear what you mean. F.i. you write: However, recurrent severe bacterial cystitis 34 occurred and neurological signs recurred with tapering of prednisolone.

What you actually are trying to say is that due to the fact that the dog developed a bacterial cystitis you needed to lower the prednisolone dose. Sadly, enough the clinical signs started to reoccur, and a different treatment plan had to be created, etc etc

3. In your introduction you are focusing on MUO. Please add the reference mentioned above and:

·      Brady SL, Woodward AP, le Chevoir M. Survival time and relapse in dogs with meningoencephalomyelitis of unknown origin treated with prednisolone and ciclosporin: a retrospective study. Aust Vet J. 2020 Oct;98(10):491-498. doi: 10.1111/avj.12994. Epub 2020 Aug 13. PMID: 32794230

·      Barber R, Downey Koos L. Treatment With Cytarabine at Initiation of Therapy With Cyclosporine and Glucocorticoids for Dogs With Meningoencephalomyelitis of Unknown Origin Is Not Associated With Improved Outcomes. Front Vet Sci. 2022 Jun 10;9:925774. doi: 10.3389/fvets.2022.925774. PMID: 35754543; PMCID: PMC9226772.

·      Lawn RW, Harcourt-Brown TR. Risk factors for early death or euthanasia within 100 days of diagnosis in dogs with meningoencephalitis of unknown origin. Vet J. 2022 Sep;287:105884. doi: 10.1016/j.tvjl.2022.105884. Epub 2022 Aug 17. PMID: 35987308.

I suggest that you also cite in you discussion:

·      Herzig R, Beckmann K, Körner M, Steffen F, Rohrer Bley C. A shortened whole brain radiation therapy protocol for meningoencephalitis of unknown origin in dogs. Front Vet Sci. 2023 Mar 20;10:1132736. doi: 10.3389/fvets.2023.1132736. PMID: 37020978; PMCID: PMC10069678.

·      Nessler JN, Oevermann A, Schawacht M, Gerhauser I, Spitzbarth I, Bittermann S, Steffen F, Schmidt MJ, Tipold A. Concomitant necrotizing encephalitis and granulomatous meningoencephalitis in four toy breed dogs. Front Vet Sci. 2022 Sep 1;9:957285. doi: 10.3389/fvets.2022.957285. PMID: 36118343; PMCID: PMC9477003.

4. Line 51 to 54: The antemortem tentative diagnosis is made based on a combination of signalment, physical and blood examination, neurological examination results, magnetic resonance imaging (MRI) findings and cerebrospinal fluid (CSF) analysis. I think the diagnosis is only made based on the MRI findings and CSF analysis

5. If no source of infection is detected in the CSF, the term ME of unknown etiology/origin (MUE/MUO) is used to make 55 a tentative diagnosis [1]. If no source of infection is detected in the CSF, the term ME of unknown etiology/origin (MUE/MUO) is used to make 55 a tentative diagnosis [1]. You are using a very old reference and we currently only use MUO. Not MUE.

6. Line 84. What is a 'High T2/FLAIR'? This is not a correct way to describe the used sequences. Stick to the international definitions.  What you mean to write (I think??) is that you used a high filed MRI scanner to make a T2-Flair (Hajnal JV, De Coene B, Lewis PD, et al. High signal regions in normal white matter shown by heavily T2-weighted CSF nulled IR sequences. J Comput Assist Tomogr 1992; 16:506-13).

7. Line 162-195. This is your justification to use this medication. It needs revision to make it readable. But the justification is solid.

8. The part form line 212 to 22 can be removed as it really does not add anything. The part from line 223-251 is not really adding to your n=2 study. You have two cases that responded to an alternative treatment. You describe why it could work. Next, I suggest to discuss the reference mentioned above and conclude that in light of the poor outcome of MUO this medication could be a suitable alternative and that more studies are needed to investigate it.

The English needs to be improved.

Author Response

We are very pleased to have been given another opportunity to revise our manuscript entitled “Off-label use of crisdesalazine (GedaCure) in meningoencephalitis in two dogs” for Veterinary Sciences. We want to extend our appreciation to you and the reviewers for taking the time and effort necessary to provide such insightful guidance. We have carefully considered comments offered by the reviewers. Herein, we explain how we revised the paper based on those comments and recommendations. The manuscript has certainly benefited from these revision suggestions. We look forward to working further with you and the reviewers to move this manuscript closer to publication.

Reviewer #3
1. COMMENT: Dear authors,
 The topic itself is interesting as there is constant debate on what might be the best treatment. You may have overlooked it but a recent paper address this:  Jeffery N; Granger N. New insights into the treatment of meningoencephalomyelitis of unknown origin since 2009: A review of 671 cases. Front Vet Sci. 2023, 10, 1114798. DOI:10.3389/fvets.2023.1114798
I advise to in cooperate this reference as well.
RESPONSE: Thank you for your kind suggestion, and we have added this reference in introduction section.

Remarks:
2. COMMENT: The first remark I have that you should use one abbreviation to describe your patients. They are suffering from a Meningoencephalo(myelitis) of Unknown Origin and the abbreviation is MUO. In your paper you constantly mix the abbreviations (f.i.line 28, 32). As you are talking about dogs with MUO stick to MUO.
RESPONSE: Thank you for your comment, and we agree with you. We changed all the terms to MUO in this manuscript.

3. COMMENT: Make clear what you mean. F.i. you write: However, recurrent severe bacterial cystitis 34 occurred and neurological signs recurred with tapering of prednisolone.
What you actually are trying to say is that due to the fact that the dog developed a bacterial cystitis you needed to lower the prednisolone dose. Sadly, enough the clinical signs started to reoccur, and a different treatment plan had to be created, etc etc
RESPONSE: Thank you for detailed comment. We have revised the sentence as follow: The symptoms were improving with immunosuppressive treatment. However, severe bacterial cystitis occurred and we couldn’t avoid tapering off prednisolone, however, neurological signs recurred. 
It has been described in the revision (Page 1, Line 33). We hope our approach acceptable.

4. COMMENT: In your introduction you are focusing on MUO. Please add the reference mentioned above and:
Brady SL; Woodward AP; le Chevoir M. Survival time and relapse in dogs with meningoencephalomyelitis of unknown origin treated with prednisolone and ciclosporin: a retrospective study. Aust Vet J. 2020, 98, 491-498. DOI:10.1111/avj.12994
Barber R; Downey Koos L. Treatment With Cytarabine at Initiation of Therapy With Cyclosporine and Glucocorticoids for Dogs With Meningoencephalomyelitis of Unknown Origin Is Not Associated With Improved Outcomes. Front Vet Sci. 2022, 9, 925774. DOI:10.3389/fvets.2022.925774
Lawn RW; Harcourt-Brown TR. Risk factors for early death or euthanasia within 100 days of diagnosis in dogs with meningoencephalitis of unknown origin. Vet J. 2022, 287, 105884. DOI:10.1016/j.tvjl.2022.105884
RESPONSE: Thank you for your kind suggestion, and we have added these references in the introduction section.

5. COMMENT: I suggest that you also cite in you discussion:
Herzig R; Beckmann K; Körner M; Steffen F; Rohrer Bley C. A shortened whole brain radiation therapy protocol for meningoencephalitis of unknown origin in dogs. Front Vet Sci. 2023, 10, 1132736. DOI:10.3389/fvets.2023.1132736
Nessler JN; Oevermann A; Schawacht M; Gerhauser I; Spitzbarth I; Bittermann S; Steffen F; Schmidt MJ; Tipold A. Concomitant necrotizing encephalitis and granulomatous meningoencephalitis in four toy breed dogs. Front Vet Sci. 2022, 9, 957285. DOI:10.3389/fvets.2022.957285
RESPONSE: Thank you for your kind suggestion, and we have added these references in the discussion section.

6. COMMENT: Line 51 to 54: The antemortem tentative diagnosis is made based on a combination of signalment, physical and blood examination, neurological examination results, magnetic resonance imaging (MRI) findings and cerebrospinal fluid (CSF) analysis. I think the diagnosis is only made based on the MRI findings and CSF analysis
RESPONSE: Thank you for detailed comment. We agree with you. We have revised the sentences as follow in the revision (Page 2, Line 52): The antemortem tentative diagnosis is made based on magnetic resonance imaging (MRI) findings and cerebrospinal fluid (CSF) analysis. If no source of infection is detected in the CSF, the term ME of unknown etiology/origin (MUE/MUO) is used to make a tentative diagnosis.

7. COMMENT: If no source of infection is detected in the CSF, the term ME of unknown etiology/origin (MUE/MUO) is used to make 55 a tentative diagnosis [1]. If no source of infection is detected in the CSF, the term ME of unknown etiology/origin (MUE/MUO) is used to make 55 a tentative diagnosis [1]. You are using a very old reference and we currently only use MUO. Not MUE.
RESPONSE: Thank you for detailed comment. We have added a recent references (Jeffery N; Granger N. New insights into the treatment of meningoencephalomyelitis of unknown origin since 2009: A re-view of 671 cases. Front Vet Sci. 2023, 10, 1114798) as your comment.

8. COMMENT: Line 84. What is a 'High T2/FLAIR'? This is not a correct way to describe the used sequences. Stick to the international definitions.  What you mean to write (I think??) is that you used a high filed MRI scanner to make a T2-Flair (Hajnal JV, De Coene B, Lewis PD, et al. High signal regions in normal white matter shown by heavily T2-weighted CSF nulled IR sequences. J Comput Assist Tomogr 1992; 16:506-13).
RESPONSE: Thank you for your detailed comment. We have revised the descriptions of MRI images as follow in the revision (Page 2, Line 83): Diffuse and multifocal T2/FLAIR hyperintense signals are observed in the cerebrum and hypothalamus, as well as the left brainstem bilaterally. In addition, mild contrast en-hancement was detected.

9. COMMNET: Line 162-195. This is your justification to use this medication. It needs revision to make it readable. But the justification is solid.
RESPONSE: Thank you for your detailed comment. We have revised the sentences in the revision (Page 4, Line 152): Crisdesalazine (GedaCure) was added on Day 188 with prednisolone 0.75 mg/kg q 12 h, under the expectation of achieving further anti-inflammatory effects.
We hope our approach acceptable.

10. COMMENT: The part form line 212 to 22 can be removed as it really does not add anything. The part from line 223-251 is not really adding to your n=2 study. You have two cases that responded to an alternative treatment. You describe why it could work. Next, I suggest to discuss the reference mentioned above and conclude that in light of the poor outcome of MUO this medication could be a suitable alternative and that more studies are needed to investigate it.
RESPONSE: Thank you for your valuable suggestion, and we agree with you. We have removed and revised the discussion section as your comment. 

Round 2

Reviewer 1 Report

I appreciate your adjustments. Mild editing to improve the English will benefit the manuscript. one example is using however, twice in one sentence (line 34-35). nothing major that prevents the reader understanding the manuscript

mild editing will benefit the manuscript. one example is using however, twice in one sentence (line 34-35). nothing major that prevents the reader understanding the manuscript

Author Response

1. COMMENT: I appreciate your adjustments. Mild editing to improve the English will benefit the manuscript. one example is using however, twice in one sentence (line 34-35). nothing major that prevents the reader understanding the manuscript.
RESPONSE: Thank you for your detailed comment, and we have revised the whole manuscript (proof-reading) including the sentence (line 34-35).

Reviewer 3 Report

Dear authors,

Thank you for your corrections. It has improved the first part of the manuscript but a through discussion is still lacking. Furthermore, the English can still be improved.

Major remark: It is good to know how crisdesalazin could work and what the motivation is (line 169 to 209 and 217 to 234). This is cleary your area of research but I miss a discussion and comparison with other used medications. What are the unique features of crisdesalazin compared to just prednisolone, cytosine arabinose and/or cyclosporine. I suggest that you incorporate the recent study of Beasley and Shores (Beasley MJ, Shores A. Perspectives on pharmacologic strategies in the management of meningoencephalomyelitis of unknown origin in dogs. Front Vet Sci. 2023 May 10;10:1167002. doi: 10.3389/fvets.2023.1167002.). In this study in detail, it is discussed. You need to discuss why this drug might be beneficial for these patients.

And you could also address, in a limitation part, why you only administered cytosine arabinose once instead of following the published protocols.

Minor remarks:

Line 25: levetiracetam; (remover however)

Line 34-35: twice however. Please remover one

Line 39: It is confusing that you state in the simple summary that it is a MUO (which is correct) and that you here still refer to ME.

Line 44: canine MUO patients, and found that it had, in these two patients, additional therapeutic effects on the MUO.

Line 45: three keywords are already in the title. I suggest adding at least MUO, cytosine arabinoside and cyclosporine. You want people to find your article.

Line 83: replace ‘we tentatively localized lesions in the brain’ into: we concluded that the neurolocalisation was intracranial and an MRI was performed.

Line 84: ‘are’ is present tense and I suggest that you use throughout the manuscript past tense. You are constantly mixing it.

Line 89: remove multifocal

Line 91-92: Replace: The prednisolone was tapered off using the protocol earlier described [7]. But please carefully read the recent work of Michaela Beasley and Andy Shores (2023)

Line 97: remove ‘suspected as neurological signs’. What you see are neurological signs. No need to mention this.

Line 113: this you need to discuss in the discussion part: other studies have advocated the use of higher dosages.

Line 118: replace ‘have’ by ‘had’

Line 121: showing a head tilt, a head turn, and circling to the left

Line 132: see line 83 so replace

Line 134: replace ‘is’ in ‘was’

Line 137: remove ‘However’

Line 140: remove ‘However’

Line 146: see line 91-92

Line 168 to 249: this is not a proper discussion. See first major remark

The most important one is not to mix  present tense and simply only use past tense.

Author Response

1. COMMENT: Major remark: It is good to know how crisdesalazin could work and what the motivation is (line 169 to 209 and 217 to 234). This is cleary your area of research but I miss a discussion and comparison with other used medications. What are the unique features of crisdesalazin compared to just prednisolone, cytosine arabinose and/or cyclosporine. I suggest that you incorporate the recent study of Beasley and Shores (Beasley MJ, Shores A. Perspectives on pharmacologic strategies in the management of meningoencephalomyelitis of unknown origin in dogs. Front Vet Sci. 2023 May 10;10:1167002. doi: 10.3389/fvets.2023.1167002.). In this study in detail, it is discussed. You need to discuss why this drug might be beneficial for these patients.
RESPONSE: Thank you for raising this important point. For both dogs, prednisolone and cytarabine were prescribed according to the recent study (Beasley and Shores et al., 2023). For case 1, the clinical signs were improved after adding crisdesalazine. And for case 2, the dosage of prednisolone for maintain clinical signs could be minimized after adding crisdesalazine. Also, we have added the reference according to your comment. We hope our approach acceptable.

2. COMMENT: And you could also address, in a limitation part, why you only administered cytosine arabinose once instead of following the published protocols.
RESPONSE: Thank you for your detailed comment. For both dogs, cytarabine was administered every 3 weeks according to general protocols. We have added this information in the revision. We hope our approach acceptable. 

3. COMMENT: Line 25: levetiracetam; (remover however)
Line 34-35: twice however. Please remover one
Line 39: It is confusing that you state in the simple summary that it is a MUO (which is correct) and that you here still refer to ME.
Line 44: canine MUO patients, and found that it had, in these two patients, additional therapeutic effects on the MUO.
Line 45: three keywords are already in the title. I suggest adding at least MUO, cytosine arabinoside and cyclosporine. You want people to find your article.
Line 83: replace ‘we tentatively localized lesions in the brain’ into: we concluded that the neurolocalisation was intracranial and an MRI was performed.
Line 84: ‘are’ is present tense and I suggest that you use throughout the manuscript past tense. You are constantly mixing it.
Line 89: remove multifocal
Line 91-92: Replace: The prednisolone was tapered off using the protocol earlier described [7]. But please carefully read the recent work of Michaela Beasley and Andy Shores (2023)
Line 97: remove ‘suspected as neurological signs’. What you see are neurological signs. No need to mention this.
Line 113: this you need to discuss in the discussion part: other studies have advocated the use of higher dosages.
Line 118: replace ‘have’ by ‘had’
Line 121: showing a head tilt, a head turn, and circling to the left
Line 132: see line 83 so replace
Line 134: replace ‘is’ in ‘was’
Line 137: remove ‘However’
Line 140: remove ‘However’
Line 146: see line 91-92
Line 168 to 249: this is not a proper discussion. See first major remark
RESPONSE: Thank you for your detailed comments, and great suggestions. We have revised the manuscript according to your all of comments. 

Round 3

Reviewer 3 Report

Thank you for the corrections.

The English has much improved but it might be good if a native speaker reads it and corrects the last few details.